# Association between Suicide and *Toxoplasma gondii* Seropositivity

**DOI:** 10.3390/pathogens10091094

**Published:** 2021-08-27

**Authors:** Laura Alejandra Mendoza-Larios, Fernando García-Dolores, Luis Francisco Sánchez-Anguiano, Jesús Hernández-Tinoco, Cosme Alvarado-Esquivel

**Affiliations:** 1Amphitheater and Department of Education, Institute of Forensic Sciences, Mexico City 06720, Mexico; lauraa.mendoza@tsjcdmx.gob.mx (L.A.M.-L.); fernando.garcia@tsjcdmx.gob.mx (F.G.-D.); 2Biomedical Research Laboratory, Faculty of Medicine and Nutrition, Juárez University of Durango State, Durango 34000, Mexico; lfsanguiano@hotmail.com (L.F.S.-A.); jhtinoco@yahoo.com (J.H.-T.)

**Keywords:** *Toxoplasma gondii*, suicide, case–control study, seroprevalence, epidemiology

## Abstract

This study aimed to determine the association between suicide and *Toxoplasma gondii* (*T. gondii*) seropositivity. Serum samples of 89 decedents who committed suicide (cases) and 58 decedents who did not commit suicide (controls) were tested for anti-*T. gondii* IgG and IgM antibodies using enzyme-linked immunosorbent assays. Anti-*T. gondii* IgM antibodies were further detected by enzyme-linked fluorescence assay (ELFA). A total of 8 (9.0%) of the 89 cases and 6 (10.3%) of the 58 controls were positive for anti-*T. gondii* IgG antibodies (OR: 0.85; 95% CI: 0.28–2.60; *p* = 0.78). Anti-*T. gondii* IgG levels were higher than 150 IU/mL in two (2.2%) cases and in five (8.6%) controls (OR: 0.24; 95% CI: 0.04–1.30; *p* = 0.11). Anti-*T. gondii* IgM antibodies were not found in any case or control using the enzyme immunoassay and were found in only one (1.7%) control using ELFA (*p* = 0.39). Rates of IgG seropositivity and high levels of anti-*T. gondii* antibodies were similar in cases and in controls regardless of their sex or age groups. The results do not support an association between *T. gondii* seropositivity and suicide. However, the statistical power of the test was low. Further research is necessary to confirm this lack of association.

## 1. Introduction

The apicomplexan protozoan *Toxoplasma gondii* (*T. gondii*) can infect any nucleated cell in any warm-blooded animal and causes a disease called toxoplasmosis [1]. The disease is an important zoonosis with medical and veterinary importance worldwide [2]. The main routes of *T. gondii* transmission are by ingestion of tissue cysts in raw or undercooked meat of infected animals, ingestion of water of raw vegetables contaminated with *T. gondii* oocysts from cat feces, and transplacental transmission [3]. Toxoplasmosis is one of the most common infections in the world due to the lifelong persistence of *T. gondii* in a latent stage [4]. Primary infection is usually subclinical, but some patients may develop cervical lymphadenopathy or ocular disease [5]. Acute infection with *T. gondii* during pregnancy is detrimental to the developing fetus [6]. In immunocompromised patients, reactivation of the latent disease can cause life-threatening encephalitis [5]. *T. gondii* can persist in the central nervous system leading to behavioral changes of the host [7]. Subacute and chronic infections with *T. gondii* are associated with an increased risk of psychiatric diseases such as schizophrenia [8]. Seroprevalence of infection with *T. gondii* has been associated with bipolar disorder [9], mixed anxiety and depressive disorder [10], aggression and impulsivity [11], obsessive–compulsive disorder, addiction [12], mental retardation [13], and generalized anxiety disorder [14]. In addition, infection with *T. gondii* has been linked to suicide attempts. Seroprevalence of *T. gondii* infection was higher in suicide attempters than in healthy controls in studies in Korea [15] and Turkey [16]. High levels of anti-*T. gondii* IgG antibodies were associated with suicide attempts in psychiatric outpatients in Mexico [17]. There is scarce information about the link between *T. gondii* seropositivity and suicide. Therefore, this study aimed to determine the association between *T. gondii* seropositivity and suicide in decedents in a forensic setting in Mexico City.

## 2. Results

A total of 8 (9.0%) of the 89 cases and 6 (10.3%) of the 58 controls were positive for anti-*T. gondii* IgG antibodies. The frequency of anti-*T. gondii* IgG antibodies in cases was similar to that found in controls (OR: 0.85; 95% CI: 0.28–2.60; *p* = 0.78). Anti-*T. gondii* IgG levels were higher than 150 IU/mL in two (2.2%) cases and in five (8.6%) controls. The frequency of high (>150 IU/mL) anti-*T. gondii* IgG antibody levels in cases was similar to that found in controls (OR: 0.24; 95% CI: 0.04–1.30; *p* = 0.11). Anti-*T. gondii* IgM antibodies were not found in any case or control using the enzyme immunoassay, whereas anti-*T. gondii* IgM antibodies were found in only one (1.7%) control using ELFA (*p* = 0.39). Stratification by age and sex of seroprevalence of *T. gondii* infection in cases and controls is shown in Table 1.

IgG seropositivity rates were similar in cases and controls regardless of their sex or age groups. Stratification by age and sex of the frequency of high (>150 IU/mL) anti-*T. gondii* IgG antibody levels in cases and controls is shown in Table 2. No association between the rates of high levels of anti-*T. gondii* antibodies and suicide among sex or age groups was found.

## 3. Discussion

Seropositivity to *T. gondii* or high anti-*T. gondii* IgG antibody levels have been associated with suicide attempts in several studies [15,16,17]; however, the link between suicide and *T. gondii* seroprevalence and serointensity has been poorly studied. Therefore, in this case–control study, we sought to determine the association between *T. gondii* seropositivity and serointensity and suicide in a forensic setting in Mexico City. We found a comparable seroprevalence of *T. gondii* infection in decedents who died by suicide and in decedents who did not die by suicide. In addition, we found a similar frequency of high levels of anti-*T. gondii* IgG antibodies in cases and in controls. Furthermore, the frequency of anti-*T. gondii* IgM antibodies in cases was similar to that found in controls. Thus, the results suggest that seropositivity and serointensity of *T. gondii* was not associated with suicide in the decedents studied in Mexico City. Our results conflict with those found in a study of women in 20 European countries where researchers identified statistically significant relationships between *T. gondii* seropositivity and suicide rates in women of postmenopausal age [18]. Our results also conflict with those found in a study of decedents in Poland where investigators observed a significantly higher seroprevalence (71.4%) of *T. gondii* infection in decedents 38–58 years old whose deaths resulted from suicide compared to the seroprevalence (44.4%) found in control groups [19]. Differences in the findings between the studies might be due to differences in the design of the studies and the study populations. In the current study, male and female decedents were included in the study, whereas only women were included in the study of 20 European countries [18]. Our study was performed in a low-*T. gondii*-seroprevalence population, whereas the study in Poland was performed in a high-*T. gondii*-seroprevalence population [19]. Mexico City (formerly called the Federal District) has an intermediate (31.0%) seroprevalence of *T. gondii* infection compared to other Mexican states, where the lowest (17.1%) seroprevalence is in the state of Baja California Sur and the highest (67.5%) seroprevalence is in the state of Tabasco using the indirect immunofluorescence assay [20]. It is unclear why the population of Mexico City has a relatively low seroprevalence of *T. gondii* infection. Studies are necessary to determine the risk factors associated with *T. gondii* infection in population groups in Mexico City. On the other hand, our results conflict with those reported in some studies of living populations where suicide attempts have been associated with *T. gondii* seropositivity or serointensity. For instance, *T. gondii* seropositivity was associated with suicide attempts in psychiatric patients in Korea [15], and *T. gondii* serointensity was associated with suicide attempts in patients with recurrent mood disorders in the USA [21], and psychiatric patients in Mexico [17]. In contrast, our results agree with a lack of association between *T. gondii* exposure and suicidal ideation found in a study of patients suffering from mental and behavioral disorders due to psychoactive substance use in Mexico [22], and a lack of association between *T. gondii* seropositivity and suicide attempts in adolescents in Turkey [23]. In addition, in a longitudinal general population study in Finland, suicidal outcomes were not associated with *T. gondii* seropositivity [24]. The negative result found in the present study does not indicate that there is not any association between *T. gondii* infection and completed suicide. Firstly, seropositivity does not indicate that *T. gondii* has infected the brain and has led to behavioral changes including suicidal behavior. Secondly, toxoplasmosis has been involved not only in suicidal behavior, but also in a number of diseases that could be present in cases as well as in controls; for instance, *T. gondii* infection has been linked to heart disease [25], and heart disease was present in some controls. In addition, the real statistical power of the test was low. The low sample size, and low statistical power of the test, increased the likelihood of a type II error. Therefore, our study may not be able to detect a true effect. Stratification by sex and age groups did not show a difference in *T. gondii* seropositivity or serointensity between cases and controls. It raises the question of whether factors other than sex and age might be involved in a link between suicide and *T. gondii* infection. It is unknown whether decedents included in the control group had had suicidal behavior in life. Recent studies in Mexico have shown an association between *T. gondii* seropositivity and suicidal behavior in several population groups including patients attending primary health care clinics [26] and people with alcohol consumption [27]. Further studies including a wide range of sociodemographic, clinical, and behavioral variables to determine the association between *T. gondii* infection and suicide should be conducted.

In the present study, the number of controls was smaller than the number of cases. Decedents who died by traffic accidents were not enrolled as controls in the study because this factor has been linked to *T. gondii* exposure [28]. The establishment of an exclusion criterion in controls but not in cases reduced the availability of controls. Results found in the 58 controls made us understand that even increasing the number of controls to 89 (the same as the number of cases) would not have changed the lack of a positive association between *T. gondii* seropositivity and suicide.

## 4. Materials and Methods

Through a case–control study design, we studied decedents who died because of suicide (cases) and decedents who died by causes other than suicide (controls). Cases and controls were studied at a forensic setting (Instituto de Ciencias Forenses) in Mexico City, Mexico, from November 2015 to December 2016. Inclusion criteria for cases were decedents who died by suicide, male or female, and of any age. Inclusion criteria for controls were decedents who died by causes other than suicide, male or female, and of comparable age to cases. In total, 89 cases and 58 controls were included in the study. Of the cases, 68 were men and 21 were women, and their mean age was 35.21 ± 17.48 (range 10–90) years. Of the controls, 48 were men and 10 were women, and their mean age was 31.82 ± 15.01 (range 8–75) years. There was no difference in age (*p* = 0.24) or gender (*p* = 0.35) between cases and controls. Data about age, sex, and cause of death of decedents were obtained from records. Causes of death in control decedents included alcohol poisoning, asphyxia, burns, carbon monoxide poisoning, electrocution, gas embolism, generalized visceral congestion, gunshot wounds, heart infarction, hypoglycemia, injuries, knife wounds, meningitis, pneumonia, pulmonary edema, strangulation, suffocation, and traumatic brain injury. The exclusion criterion for controls was decedents who died by traffic accidents.

Blood samples of decedents were collected more than 12 h after death. After blood centrifugation, serum was obtained and frozen down at −20 °C, then analyzed. All serum samples were analyzed for anti-*T. gondii* IgG and IgM antibodies using the commercially available enzyme immunoassays “*Toxoplasma gondii* IgG” kit (Diagnostic Automation/Cortez Diagnostics, Inc., Woodland Hills, CA. USA) and “*Toxoplasma gondii* IgM” kit (Diagnostic Automation/Cortez Diagnostics, Inc.), respectively. Anti-*T. gondii* IgG antibody levels were expressed as International Units (IU)/mL, and *a* value of 8 IU/mL was used as a cut-off for seropositivity. Anti-*T. gondii* IgG antibody levels were arbitrarily considered as “high” when found at >150 IU/mL. We selected this high antibody level because the highest calibrator provided in the commercially available enzyme immunoassay used was 150 IU/mL and for comparison purposes with other studies in the region that have used this high antibody level. In addition, anti-*T. gondii* IgM antibodies were further detected using the commercially available enzyme-linked fluorescent assay (ELFA) kit “VIDAS Toxo IgM” (BioMérieux, Marcy-l’Etoile, France). All assays were performed according to the instructions of the manufacturers.

Statistical analysis was performed with the aid of the software Epi Info version 7 and Microsoft Excel. We calculated the sample size using the following parameters: a two-sided confidence level of 95%, a power of 80%, a 0.65 ratio of controls to cases, a reference seroprevalence of 27.97% [29] as the percentage outcome of exposure in controls, and an odds ratio of 2.7. The result of the sample size calculation with the Fleiss method was 88 cases and 57 controls. The Student’s t-test was used to compare the age of cases and controls. The seropositivity rates between the groups were compared with the Pearson’s Chi-square test or the Fisher exact test (when *a* value was ≤5). We calculated the odd ratios (OR) and 95% confidence intervals (CI), and a *p* value less than 0.05 was considered statistically significant. An additional statistical analysis to determine the statistical power of the test was performed. For this purpose, we used the following data: a sample size of 89 cases and 58 controls, a 10.3% seroprevalence of *T. gondii* infection, an OR of 2.7, and a significance level of 0.05. The actual power of the test was 53.5%.

## 5. Conclusions

The results of this study of decedents in Mexico City do not support an association between *T. gondii* seropositivity and suicide. Further research is necessary to confirm this lack of association.

## Figures and Tables

**Table 1 pathogens-10-01094-t001:** Association between suicide and *T. gondii* seropositivity: stratification by sex and age groups.

	Cases			Controls					
		Seropositivity			Seropositivity			95%	
	No.	To *T. gondii*		No.	To *T. gondii*			Confidence	*p*
Characteristic	Tested	No.	%	Tested	No.	%	OR	Interval	Value
Sex									
Male	68	6	8.8	48	6	12.5	0.67	0.20–2.24	0.52
Female	21	2	9.5	10	0	0.0	-	-	1.00
Age (years)									
≤30	43	6	14.0	35	3	8.6	1.72	0.39–7.48	0.50
31–50	30	1	3.3	15	1	6.7	0.48	0.02–8.29	1.00
>50	16	1	6.3	8	2	25.0	0.20	0.01–2.64	0.24
All	89	8	9.0	58	6	10.3	0.85	0.28–2.60	0.78

**Table 2 pathogens-10-01094-t002:** Association between suicide and high (>150 IU/mL) levels of anti-*T. gondii* IgG: stratification by sex and age.

	Cases			Controls					
		>150 IU/mL			>150 IU/mL			95%	
	No.	Of IgG		No.	Of IgG			Confidence	*p*
Characteristic	Tested	No.	%	Tested	No.	%	OR	Interval	Value
Sex									
Male	68	1	1.5	48	5	10.4	0.12	0.01–1.13	0.08
Female	21	1	4.8	10	0	0.0	-	-	1.00
Age (years)									
≤30	43	2	4.7	35	2	5.7	0.80	0.10–6.02	1.00
31–50	30	0	0.0	15	1	6.7	-	-	0.33
>50	16	0	0.0	8	2	25.0	-	-	0.10
All	89	2	2.2	58	5	8.6	0.24	0.04–1.30	0.11

## Data Availability

Data are provided within the article.

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
