# Peer review of "Association between Suicide and Toxoplasma gondii Seropositivity"

_pathogens, 2021, doi:10.3390/pathogens10091094_

Round 1

Reviewer 1 Report

The manuscript is a case control study on the prevalence of IgG and IgM antibodies to Toxoplasma gondii and the relationship with completed suicide versus people who died due to different causes, excluding traffic accidents, because traffic accidents have been linked to T. gondii infection in some studies. The study is well conceived with homogenous age and sex proportions in both control and suicide groups for comparison. Although the authors mention that the number of controls is a limitation, 89 suicide cases compared to 59 controls seems adequate for comparison purposes.

The cut-off for positivity is 8 IU/ml, how did the authors select the high-level antibodies at 150 IU/ml?

There is a mistake in line 159. Instead of “we actual power”. It should say “the actual power”.

The authors mention there is a low seroprevalence population in the area studied in Mexico. Could they reference the values in the area of study versus other areas of Mexico? Which would be the reason/s for the low T. gondii seroprevalence in the study area?

When was the blood samples collected? Just after the suicide or death or later on?

Author Response

  1. The cut-off for positivity is 8 IU/ml, how did the authors select the high-level antibodies at 150 IU/ml?

Anti-T. gondii IgG antibody levels were arbitrarily considered as “high” when found at >150 IU/ml.  We selected this high antibody level because the highest calibrator provided in the commercially available enzyme immunoassay used was 150 IU/ml, and for comparison purposes with other studies in the region that have used this high antibody level (lines 159-163).

  1. There is a mistake in line 159. Instead of “we actual power”. It should say “the actual power”.

Corrected. The word “We” was replaced with “The” (line 179).

  1. The authors mention there is a low seroprevalence population in the area studied in Mexico. Could they reference the values in the area of study versus other areas of Mexico? Which would be the reason/s for the low T. gondii seroprevalence in the study area?

We added the following information concerning the seroprevalence of T. gondii infection in Mexico (lines 91-97):

Mexico City (formerly called Federal District) has an intermediate (31.0%) seroprevalence of T. gondii infection as compared with other Mexican states, having the lowest (17.1%) seroprevalence the state of Baja California Sur and the highest (67.5%) seroprevalence the state of Tabasco using the indirect immunofluorescence assay [20]. It is unclear why the population of Mexico City has a relatively low seroprevalence of T. gondii infection. Studies to determine the risk factors associated with T. gondii infection in population groups in Mexico City are needed.

  1. When was the blood samples collected? Just after the suicide or death or later on?

Blood samples were collected more than 12 hours after death (line 152).

Thank you for your valuable comments for improving our manuscript.

Reviewer 2 Report

The manuscript „Association between suicide and Toxoplasma gondii seropositivity” described the the association between suicide and Toxoplasma gondii seropositivity.

The study were conducted among decedents who committed suicide and decedents who did not commit suicide (controls) with using 2 immunoassays.

I have one comment for the authors of manuscript.

Discussion section: please expand this part with other results from the other study, especially that you obtained in both tables in the Results parts some interesting data.

Author Response

  1. Discussion section: please expand this part with other results from the other study, especially that you obtained in both tables in the Results parts some interesting data.

We expanded the Discussion section by adding the following information (lines 91-97, and 117-126):

Mexico City (formerly called Federal District) has an intermediate (31.0%) seroprevalence of T. gondii infection as compared with other Mexican states, having the lowest (17.1%) seroprevalence the state of Baja California Sur and the highest (67.5%) seroprevalence the state of Tabasco using the indirect immunofluorescence assay [20]. It is unclear why the population of Mexico City has a relatively low seroprevalence of T. gondii infection. Studies to determine the risk factors associated with T. gondii infection in population groups in Mexico City are needed.

Stratification by sex and age groups did not show a difference in T. gondii seropositivity or serointensity between cases and controls. It raises the question whether factors other than sex and age might be involved in a link between suicide and T. gondii infection. It is un-known whether decedents included in the control group had had suicide behavior in life. Recent studies in Mexico have shown an association between T. gondii seropositivity and suicide behavior in several population groups including patients attending primary health care clinics [26], and people with alcohol consumption [27]. Further studies including a wide range of sociodemographic, clinical, and behavioral variables to determine the association between T. gondii infection and suicide should be conducted.

Thank you for your valuable comments for improving our manuscript.

Round 2

Reviewer 2 Report

Thank you for your response and correction.

In my opinion the manuscript in the present form form is acceptable.

This manuscript is a resubmission of an earlier submission. The following is a list of the peer review reports and author responses from that submission.

Round 1

Reviewer 1 Report

This manuscript reports no association between toxoplasmosis and suicide among sex or age groups. In my feedback below, I offered several suggestions that I believe will add to the strength of this manuscript.

It is necessary to add information about the characteristics of the control sample. The authors only provide information that the criteria for inclusion in the control group were decedents who died by causes other than suicide, male or female, and at a comparable age to the cases. What were the causes of death in the control group? It would be helpful to add this information.

If the causes of death of controls were various diseases, I recommend adding this information to the discussion. Toxoplasmosis has repeatedly been positively correlated with many diseases, such as cardiovascular diseases. Therefore, it is appropriate to consider the reported negative results as a possible influence of toxoplasmosis not only on suicides but also on other causes of deaths in control subjects.

A total of 89 cases and 58 controls were included in the study. Why are there fewer controls than cases in the study? Please, provide more information about collecting controls.

The authors concluded that the results of their study of decedents in Mexico City do not support an association between T. gondii seropositivity and suicide. This conclusion is subject to Type II error as there is a limited number of positive samples. There is not enough power to detect any significant relationship. Based on the results of power analyses, the authors could estimate how many subjects would be needed to demonstrate the effect of toxoplasmosis on suicide.

Author Response

  1. It is necessary to add information about the characteristics of the control sample. The authors only provide information that the criteria for inclusion in the control group were decedents who died by causes other than suicide, male or female, and at a comparable age to the cases. What were the causes of death in the control group? It would be helpful to add this information.

Information about the causes of death in the control group was added (lines 132-136).

  1. If the causes of death of controls were various diseases, I recommend adding this information to the discussion. Toxoplasmosis has repeatedly been positively correlated with many diseases, such as cardiovascular diseases. Therefore, it is appropriate to consider the reported negative results as a possible influence of toxoplasmosis not only on suicides but also on other causes of deaths in control subjects.

Further discussion about the negative results was added (lines 104-111).

  1. A total of 89 cases and 58 controls were included in the study. Why are there fewer controls than cases in the study? Please, provide more information about collecting controls.

Decedents who died by traffic accidents were not enrolled as controls in the study. The presence of an exclusion criterium in controls but not in cases reduced the availability of controls (lines 112-115).

Results found in the 58 controls provided us information to understand that even by increasing the number of controls to 89 (as the number of cases) would not change the lack of a positive association between T. gondii seropositivity and suicide (lines 112-118).   

  1. The authors concluded that the results of their study of decedents in Mexico City do not support an association between T. gondii seropositivity and suicide. This conclusion is subject to Type II error as there is a limited number of positive samples. There is not enough power to detect any significant relationship. Based on the results of power analyses, the authors could estimate how many subjects would be needed to demonstrate the effect of toxoplasmosis on suicide.

A sample size calculation using a power of 80% was performed. The result of the sample size calculation using the Fleiss methods was 88 cases and 55 controls. Information about the sample size calculation including the power was provided in the Materials and methods section (lines 149-153).   

Thank you for your valuable comments for improving our manuscript.

Reviewer 2 Report

The MS presented for review is related to an attempt to obtain an answer to the question, whether Toxoplasma gondii infection leads to more risky behavior, in this case even more drastic behavior, such as suicide.

This is a typical case study with very limited sample size, and on this basis, the authors try to answer to one of the most interesting questions related to T. gondii infections.

The main conclusion presented by the authors is: “Results of this study of decedents in Mexico City do not support an association be-138 tween T. gondii seropositivity and suicide.” However, the authors do not seem to notice that the research was conducted on a very small sample (N = 89), while seropositivity was noted in eight cases.

Along a such small sample size, creating categories, such as age, gender, etc., cause an even greater dispersion of the very limited data. In my opinion, the results presented in this MS do not bring anything new to the knowledge on the relationship between risky behavior and T. gondii infection. Moreover, the publication of this data will force the authors of subsequent studies to cite this work. In my opinion, this study should be published (if ever) in some local journal, but it is not suitable as a publication in an international journal with IF higher than 3.

Author Response

The main conclusion presented by the authors is: “Results of this study of decedents in Mexico City do not support an association be-138 tween T. gondii seropositivity and suicide.” However, the authors do not seem to notice that the research was conducted on a very small sample (N = 89), while seropositivity was noted in eight cases.

A sample size calculation using a power of 80% was performed. The result of the sample size calculation using the Fleiss methods was 88 cases and 55 controls. Information about the sample size calculation including the power was provided in the Materials and methods section (lines 149-153).  

Along a such small sample size, creating categories, such as age, gender, etc., cause an even greater dispersion of the very limited data. In my opinion, the results presented in this MS do not bring anything new to the knowledge on the relationship between risky behavior and T. gondii infection. Moreover, the publication of this data will force the authors of subsequent studies to cite this work. In my opinion, this study should be published (if ever) in some local journal, but it is not suitable as a publication in an international journal with IF higher than 3.

The sample size of cases and controls used in the present study is supported by a sample size calculation (lines 149-153).

Results found in the 58 controls provided us information to understand that even by increasing the number of controls to 89 (as the number of cases) would not change the lack of a positive association between T. gondii seropositivity and suicide (lines 115-118).   

We are optimistic that our findings can be cited in more than 3 studies.

Our results can be useful for a meta-analysis too.

Thank you for your valuable comments for improving our manuscript.

Round 2

Reviewer 2 Report

Due to the fact that the MS was not corrected, because it could not be improved, my opinion on this matter did not change. In my opinion, the manuscript should not be accepted for publication.